# Utilization of Genetic Resources, Genetic Diversity and Genetic Variability for Selecting New Restorer Lines of Rice (*Oryza sativa* L.)

**DOI:** 10.3390/genes13122227

**Published:** 2022-11-27

**Authors:** Mamdouh M. A. Awad-Allah, Wafaa W. M. Shafie, Moodi Saham Alsubeie, Aishah Alatawi, Fatmah Ahmed Safhi, Salha Mesfer ALshamrani, Doha A. Albalawi, Hadba Al-Amrah, Dikhnah Alshehri, Rana M. Alshegaihi, Mohammed A. Basahi, Abdurrahman S. Masrahi

**Affiliations:** 1Rice Research Department, Field Crops Research Institute, Agricultural Research Center, Giza 12619, Egypt; 2Central Laboratory for Design and Statistical Analysis Agricultural Research Center, Giza 12619, Egypt; 3Biology Department, College of Science, Imam Mohammad Ibn Saud Islamic University (IMSIU), Riyadh 11623, Saudi Arabia; 4Department of Biology, Faculty of Science, University of Tabuk, Tabuk 71421, Saudi Arabia; 5Department of Biology, College of Science, Princess Nourah bint Abdulrahman University, P.O. Box 84428, Riyadh 11671, Saudi Arabia; 6Department of Biology, College of Science, University of Jeddah, Jeddah 2795, Saudi Arabia; 7Department of Biology, Faculty of Science, University of Tabuk, Tabuk 71491, Saudi Arabia; 8Department of Biological Sciences, Faculty of Science, King Abdulaziz University, Jeddah 21589, Saudi Arabia; 9Department of Biology, College of Science, University of Jeddah, Jeddah 21493, Saudi Arabia; 10College of Science and Arts Sajir, Shaqra University, P.O. Box 33, Shaqra 11961, Saudi Arabia; 11Biology Department, Faculty of Science, Jazan University, Jazan 45142, Saudi Arabia

**Keywords:** rice, restorer lines, genetic variability, cluster analysis, principal component analysis, genetic divergence

## Abstract

Exploration of and understanding diversity and variability in genotypes of germplasm determines the success of rice improvement programs. One of the most important determinants of the success of breeding programs is genetic diversity and inheritance of traits. Genetic variability analysis helps breeders to determine the appropriate selection method and standards to be used to improve the preferred trait. The aim of this study was to estimate genetic components, heritability and to obtain information about genetic diversity using cluster analysis and principal component analysis. Twenty rice genotypes with three replicates in a randomized complete block design were analyzed at the Experimental Farm at Sakha Agricultural Research Station, Sakha, Kafr El-Sheikh, Egypt, during the period from 2017 to 2020. The results of the analysis of variance showed that highly significant variations were recorded between the studied genotypes for all traits. The genotypic coefficient of variation (GCV%) and phenotypic (PCV%) coefficient of variation were moderate for plant height, panicles/plant, panicle weight, spikelets/panicle, filled grains/panicle, grain yield/plant and amylose content percentage for the first-year, second-year and combined data. This indicates the existence of beneficial genetic variability that can be exploited to improve these traits. The broad-sense estimates of heritability were high and recorded values higher than 60% for all the studied traits for the two-year and combined data, except for hulling percentage. This indicates that the selection of traits that have high heritability and are less affected by the environment leads to an acceleration of the improvement of these traits. The results from the cluster analysis and principal component analysis revealed a high level of genotypic variation among the studied genotypes and genetic diversity between them. One of the most important outcomes of this study is the successful utilization of genetic resources (germplasm) from ancient varieties and lines of rice in selecting and identifying 17 new restoration lines of rice, which have various improvement purposes in rice and hybrid rice breeding programs.

## 1. Introduction

Rice is the main food crop for above half of the population of the world. In Egypt, rice is the second most important cereal crop in terms of nutrition and productivity after wheat [1,2,3,4]. To meet the increasing demand for rice as a result of the population increase and in light of the decrease in arable land, the decrease in irrigation water and the decrease in chemical fertilizers, it is necessary to increase the production capacity of rice, and to achieve this, we need to use several different methods and strategies. First, we need rice varieties that are characterized by high productivity with great stability in this area. These methods and strategies also include the breeding of hybrid rice, which can increase production and yield by 15–20% compared to the best inbred rice varieties [3,5]. By introducing and expanding the cultivation of hybrid rice, this will help to overcome the gaps in the rice yield and overcome the difficulties associated with increasing rice production and preserving the available natural resources [6]. Many countries around the world depend on the use of a three-line system to breed and produce most types of hybrid rice, which is based on the use of cytoplasmic genetic male sterile lines [7,8]. By following this system, effective restoration lines are required in order for the hybrid rice breeding programs to succeed in developing new types of hybrid rice [9]. Therefore, one of the vital factors for the success of these breeding programs is the use and adoption of the correct method for the selection of the parents involved in hybridization [10]. One of the most important issues in the breeding of hybrid rice in Egypt is the low number of restore lines, as there is only one commercial restore line available for CMS-line rice in Egypt [1,11,12].

The level of heterosis in several hybrid rice varieties is reported to be the highest in indica × japonica hybrids [13]. The hybrid vigor of lines with wide genetic divergence increases the yield and crop production. Genetic distance is significantly and positively related to heterosis in rice and this has been reported by several studies, which have also reported that increased yield is more heterogeneous in hybrids [14,15]. The increase in yield of indica–indica rice hybrids was higher than 15% compared to the best pure indica varieties, while the yield of hybrids between Indica and Japonica increased by 25% compared to the best pure indica varieties [16]. 

Amylose content is the most important characteristics of rice grain quality and is directly related to cooking and eating quality [17]. However, the required quality may differ from one geographical area to another, which depends on the preference of the consumer [18,19]. 

The improvement of agronomic traits helps to increase crop productivity in breeding programs. In the development of new high-yielding varieties, genetic resources play an important role [20].

Knowledge on heritability and genetic advance is fundamental to identify the crop characteristics amenable to genetic improvement through selection. It is important to emphasize that heritability would not be practically profitable without the consideration of genetic advance in breeding programs based on phenotypic selection [21]. Improvement of a quantitative and low heritable characteristic such as grain yield may be more successful and faster through the selection of its more heritable components that show evidence of significant positive association.

Genetic diversity is considered as a major component of germplasm and is important for breeding and improving rice lines and varieties to meet the current and future food requirements [22,23]. To expand the genetic base of a breeding program, it is necessary to use a population with a high level of genetic variance [24]. Furthermore, several researchers have reported that knowledge on genetic diversity provides key information for both basic and applied studies [22].

To explore and study the variability present in rice germplasm in order to determine the desired agronomic traits, genetic diversity analysis is used. The International Rice Research Institute (IRRI) Germplasm Center preserves the world’s rice germplasm, conserving about 100,000 genotypes out of a total of 140,000 diverse genotypes of rice across the world [25]. These genotypes have many desirable traits, which can be exploited using genetic enhancement and in various breeding and genetic improvement programs. Determination of rice genotypes with restorer fertility ability and with a high yield with stable yield is an important primary objective in the identification and breeding of new and good parental lines of hybrid rice as well as new promising hybrids of rice. To determine the degree of genetic divergence between the tested genotypes based on their performance and contributing characteristics, cluster analysis is used as an effective tool to achieve this purpose. Principal component analysis (PCA) and cluster analysis based on phenotypic attributes can be used to assess genetic variability [26,27].

Principal component analysis (PCA) is an effective approach to measure genetic divergence between germplasm genotypes with respect to their characteristics [28]. Cluster analysis is also an effective method for selecting parents to achieve high hybrid vigor to strengthen and support hybrid rice breeding programs. With this background, the current study was conducted to understand and categorize the magnitude and nature of genetic diversity among the studied germplasm lines and genotypes.

Genotype x trait (GT) biplot analysis allows and helps us to visualize the true correlation between traits and understand the relationships that facilitate the identification of traits that can be used in indirect selection to improve the grain yield [29,30,31]. In addition, GT biplot analysis provides information about the usefulness of genotypes and lines for productivity, as well as information on less important (redundant) traits. Swelam (2012) [32] used a GT biplot graph to visualize the relationships among genotypes and their traits. They stated that the grain yield of wheat was closely correlated with grains/spike, grain weight and spike length. Therefore, GT biplot graphs can be successfully used for multi-trait selection in wheat breeding programs. Despite the recent interest shown in GT biplot graphs as a tool that can be used to interpret the two-way tables of genotypes and traits, they are rarely used in the yield trials in Egypt. Few studies were found concerning this technique.

Therefore, the present study aimed to select and evaluate some genotypes of rice to identify new restorer lines and these restorer lines may be used to develop rice hybrids. The objectives of this work are as follows: (1) to estimate broad sense heritability (hb2) and genetic advance (% mean) for yield and its attributes, (2) to investigate genetic diversity among the tested genotypes using cluster analysis, (3) to identify genotype and trait relationships using GT biplot graphs; (4) to discuss whether GT biplot graphs are a good alternative tool to be used alongside and cluster analysis.

## 2. Materials and Methods

### 2.1. Plant Materials

Twenty rice genotypes were selected from a core collection sourced from several different countries and arranged in a randomized block design with three replications, as shown in Table 1. These genotypes included restorer lines and two CMS lines from a previous test cross experiment that involved the hybridization of these two CMS lines, which were identified as parental lines in the hybrid rice breeding program. Each replication contained 20 genotypes that were randomized and replicated within each block. At the age of twenty-five days of the seedlings, the genotypes were transplanted and placed 20 cm inside the row and 20 cm between rows. Recommended cultural practices were applied.

### 2.2. Field Evaluation

In the laboratories of the rice research department, and the Farm of Sakha Agricultural Research Station, Sakha, Kafr El-Sheikh, Egypt, during the period from 2017 to 2020, the evaluation of the experiments of this study was carried out.

### 2.3. Data Collection

From within the central rows for each replicate, five plants were randomly selected to evaluate the grain yield and its components, as well as three of the grain quality characteristics.

The following characteristics were studied: number of days to 50% heading (day) (DH), plant height (cm) (PH), panicles/plant (P/P), panicle length (cm) (PL), pollen fertility (%) (PF%), spikelets/panicle (Sp/P), spikelet fertility (%) (SF%), filled grains/panicle (Fg/P), panicle weight (g) (PW), 1000 grain weight (g) (GW), grain yield/plant (g) (GY/P), hulling percentage (H%), milling percentage (M%), and amylose content percentage (AC%).

The IRRI Standard Evaluation System for rice was followed during the data collection, recording of all measurements and estimation of characteristics [33].

Parental lines were classified as maintainers, restorers, partial maintainers and partial restorers using the scale adopted by Virmani et al. (1997) [7]. 

Estimation of hulling and milling %: 

The calculation of percentage of hulling recovery and milling recovery was done with the formula mentioned by Xangsayasane et al (2009) [34].

Determination of amylose content:

The method proposed by Juliano in 1971 [35] was followed and used to determine the amylose content in milled rice.

### 2.4. Statistical Analysis

The data collected were analyzed using an individual and combined analysis of variance (ANOVA) to design randomized complete blocks for each year and over the two years [36]. Bartlett’s test was carried out before the combined analysis to test the homogeneity of individual error terms [37]. A least significant difference (LSD) test was used to detect the significant and highly significant differences among genotype means at 0.05 and 0.01 probability levels.

The genotypic and phenotypic variations and their corresponding coefficient of variations were estimated using the pertinent mean square expectations according to the method suggested by Johnson et al. in 1955 [38,39]. Broad sense heritability (hb2) and genetic advance in terms of percentage of means (with 5% selection intensity) were estimated as described by Allard (1999) [40].

The values of GCV% and PCV% were classified into three categories according to [41], which were as follows: low = 0–10%, medium = 10–20% and high => 20%.

The h2bs estimates were also categorized according to the proposal by [38] and were as follows: low = 0–30%; medium = 30–60%, high = above 60%.

A scale of Euclidean distance and a minimal variance wing method, as defined by Ward (1963) [42], were used to perform a hierarchical cluster analysis on the standardized data. 

The genotype x trait (GT) biplot method was used, as demonstrated in the work of Yan and Rajcan (2002) [29]. Since the traits were measured in different units, a biplot was constructed using standardized values of the traits’ means.

## 3. Results

### 3.1. Test Cross Experiment

The new hybrid combinations showed a pollen fertility percentage of more than 85%; the hybrids IR79156A × G11, IR79156A × G7, G46A × G11, IR79156A × G5, IR79156A × G9, IR79156A × G8, G46A × G4 and G46A × G16 showed the highest values, as shown in Appendix A. On the other hand, the spikelet fertility percentage of the new hybrid combinations tested was more than 75% and the hybrids IR79156A × G11, IR79156A × G7, IR79156A × G5, IR79156A × G9, IR79156A × G8, G46A × G11, G46A × G16, G46A × G4 and G46A × G4 showed the highest values, as shown in Appendix A.

### 3.2. Mean Performance

The data tabulated in Appendix A show the mean values of grain yield and yield components for the first year, second year and over the two years. For number of days to 50% heading, most of the studied genotypes showed low values. The genotypes G19, G13 and G7 recorded the lowest desirable values for the two-year and the combined data over two years, with the values 79.7, 101.2 and 103.1 days over the two years, as shown in Appendix A and Figure 1. The data in Appendix A and Figure 2 show that the genotypes G19, G1, G14 and G4 recorded the lowest values of plant height, with the values 88.84, 94.07, 96.03, 96.13, 96.13, 100.97, 102.72 and 103.83, respectively, while the genotypes G6, G8, G17, G18 and G20 showed the highest desirable values for the combined data over two years. 

On the contrary, the genotypes G14, G17 and G18 recorded the highest numbers of panicles/plant for both years and the combination of the two years, as shown in Appendix A and Figure 3. 

The genotypes G18, G13, G10, G11 and G12 recorded the highest values for the first-year and second-year data, as well as the combined data, for panicle length, as shown in Appendix A. 

Concerning panicle weight, 7 out of 21 studied genotypes recorded values higher than G1 (check variety). The genotypes G9, G13, G3, G11, G18, G16 and G7 demonstrated the highest mean values for the two-year and combined data, respectively, as shown in Appendix A and Figure 4. 

However, the genotypes G1, G2, G4, G6, G17, G9 and G3 recorded the highest values of pollen fertility % and spikelet fertility %, as shown in Appendix A. The data of the combined analysis over the two years demonstrated that the genotypes G11, G7, G20 and G9 gave the highest numbers of spikelets/panicle and numbers of filled grains/panicle, as shown in Appendix A and Figure 5 and Figure 6.

All the values of 1000-grain weight for the studied genotypes were higher than to it for G1 (the only commercially available restorer line) for the two-year and the combined data, as shown in Appendix A and Figure 7. The genotypes G13, G11, G17, G15, G16, and G8 gave the highest values of grain yield/plant for both years and for the combination of both years, as shown in Appendix A and Figure 8. 

Appendix A and Figure 9 reveal the data of the traits hulling %, milling % and amylose content % for the two-year and combined data. The genotypes G7, G15, G1, G12 and G10 showed the highest values of hulling percentage for the two-year and combined data, respectively, as shown in Appendix A. However, the genotypes G9, G14, G6, G17, G4, G3, G18, G15, G13, G7, and G10 recorded the highest values of milling percentage, as shown in Appendix A and Figure 9. For amylose content percentage, the following five genotypes showed desirable values that ranged between low or intermediate values for the combined data over two years: G1 (16.52), G2 (17.87), G13 (18.04), G11 (19.63), and G8 (21.40), as shown in Appendix A. 

### 3.3. Analysis of Variances

The data in Table 2, Table 3, Table 4 and Table 5 show that the mean squares for grain yield, yield components and grain quality traits of twenty rice lines for the two-year and the combined data over the two years. The results of the analysis of variances showed the highly significant values of the mean squares of genotypes for the studied traits for the two years.

On the other hand, the results of the analysis of variance for the combined data demonstrated that there were highly significant and significant differences between the years for all the studied traits, except for panicle weight and 1000 g grain weight, as shown in Table 6, Table 7 and Table 8. Furthermore, the results of the analysis of variance for the combined data showed highly significant values for the mean squares of genotypes for all studied traits, as shown in Table 6, Table 7 and Table 8. The interaction between year and genotypes (year × genotypes) was also highly significant for all studied traits, except days to 50% heading, 1000 g grain weight, grain yield/plant, hulling %, milling % and amylose content %. 

### 3.4. Genetic Parameters

The data in Table 9 demonstrated that the phenotypic coefficients of variance were high for panicle weight (20.39, 23.12 and 20.22%), while they were moderate for plant height (12.28, 11.52 and 11.58%), number of panicles/plant (13.81, 12.21 and 14.71%), number of spikelets per panicle (16.62, 17.66 and 14.71%), number of filled grains/panicle (16.09, 17.18 and 14.59%), grain yield per plant (20.10, 17.96 and 18.98%) and amylose content percentage (15.96, 14.94 and 15.42%) for the first-year, second-year and combined data, respectively.

However, the genotypic coefficients were moderate for plant height (12.14, 11.20 and 11.44%), number of panicles/plant (12.65, 10.77 and 13.09%), panicle weight (16.99, 21.12 and 18.24%), number of spikelets per panicle (14.28, 16.18 and 13.09%), number of filled grains/panicle (13.51, 15.82 and 12.79%), grain yield per plant (18.27, 16.18 and 17.18%) and amylose content percentage (15.61, 14.60 and 15.07%) for the first-year, second-year and combined data, respectively, as shown in Table 9. 

Estimates of genetic variability indicated that the PCV values were slightly higher than those for GCV that corresponded to the traits under study. 

The results showed that the heritability values in a broad sense (H2%) were high for all the studied traits, and the recorded values were more than 60% for the studied traits, except for hulling percentage (55.59, 49.95 and 51.99%) for the two-year and combined data (Table 9). 

High heritability in a broad sense, and high genetic advance recorded as a percentage of the mean were noted for the following characteristics: PH, P/P, PW, Sp/P, Fg/P, GY/P and AC %, as shown in Table 9. 

On the other hand, high heritability and moderate genetic advance were recorded for DH, PL, and 1000 g GW. 

On the contrary, high heritability and low genetic advance were recorded for days to 50% flowering, PF%, SF%, and M%, as shown in Table 9. 

### 3.5. Cluster Analysis

Cluster analysis was performed and a dendrogram was generated based on the genetic similarity coefficient (Figure 10). A total of 20 rice genotypes based on 14 characteristics were classified into 5 group of clusters, with 3 genotypes in the first cluster (designated as C-A), 6 in C-B, 6 in C-C, 1 in C-D, and 4 in C-F, as shown in Table 10 and Figure 10. 

The lines and genotypes classified as cluster group D had higher mean values for the seven traits studied. Cluster A had lower desirable mean values regarding plant height and amylase content. The check varieties Giza 178 and Giza 181 were also included in this cluster. The lines and genotypes classified as cluster B had higher mean values of Sp/P and grain yield/plant. On the other hand, the lines and genotypes classified as cluster group F had higher mean values for three of the traits studied, including P/P, PL and H%. 

### 3.6. Principal Component Analysis (PCA)

The separation of PC 1, PC 2, PC 3 and PC 4 revealed that 11 lines were detached in quarter 1 (G9, G18, G17, G16, G3 and G15) and quarter 2 (G14, G10, G5, G6, G4 and G2). The genotypes 9, 18, 17, 16, 3 and 15 and traits such as PW, PH, PL, PP, M%, 1000-GW, AC% and DH showed more variation than any other genotype or trait and were located in quarter 1, while SF% was located in quarter 2 (Figure 11 and Table 11). 

PC1 and PC2, as the main PCs, analyzed the diversity of the agronomic characteristics of the genotypes studied. The results of the PCA of the agronomic characteristics of the genotypes studied showed that PC1 and PC2 were the best PCs in identifying the diversity between yield and its components, while PC4 was the optimal PC in identifying the diversity between grain quality traits (Figure 11 and Table 11). 

Information on all the traits in each group was collected through PCA. Based on the results of the PCA, for PC1, the traits DH, P/P, PF%, SF%, 1000-GW and M% had a higher proportion of variance. Moreover, from the results of the PCA, for PC2, it can be observed that the traits PH, PL, PW, SP/P and GY/P had a higher proportion of variance. On the contrary, the results of the PCA revealed that for PC4, the traits H%, M% and AC% had a higher proportion of variance (Figure 11 and Table 11). 

## 4. Discussion

### 4.1. Selection and Evaluation

The first step in successful rice breeding and hybrid rice breeding programs is the successful selection of parental lines, whether this is for breeding new hybrids or improving parental lines, new lines and varieties in rice, and the success of these programs depends on this step [1,3]. The studied genotypes, i.e., lines and varieties, were selected from the germplasm available for rice, then evaluated in the following year, observed, and crossed with the two most important cytoplasmic male sterile lines to set up the experimental test cross. Based on the percentage of pollen fertility and the percentage of spikelet fertility of the resulting hybrids and the results of the test cross experiment, the selected lines were divided based on the results of these two traits of the resulting hybrids and their genetic behavior into maintainer lines, partial maintainer lines, partial restorer lines and restorer lines. In the next step, the best 17 restorer fertility lines were selected for cultivation and evaluated with the two maintainer lines and the only restorer fertility commercial line. The eighteen restorer lines were also re-crossed with the same two cytoplasmic male sterile lines.

### 4.2. Test Cross Experiment

The results of the test cross experiment showed that the pollen fertility percentage of the new hybrid combinations was more than 85%. Similar results were observed by [1,43,44,45,46,47]. On the other hand, the spikelet fertility percentage of the new hybrid combinations tested was more than 75% and these results agreed with the results by [1,43,44,45,48]. These findings indicted that these lines can be identified as restorer lines. On the other hand, the restorer and maintainer lines were determined using pollen and spikelet fertility analysis in a test cross experiment, which is one of the most important steps for developing new hybrids. [1,44]. These highly fertile hybrids can be used in developing promising new hybrids [1,43,44,45,48].

### 4.3. Mean Performance

The mean results of grain yield and yield components for the first-year, second-year and combined data over two years are tabulated in Appendix A. The genotypes showed a high amount of variation among themselves for the studied traits. The values of mean performance were mostly higher in the second year than the first year and were very similar, but with slight differences. Thus, these genotypes show synchronized flowering with two CMS lines, and this synchronization is useful in the field of seed production, as it increases the rate of pollination and out-cross, and thus increases seed production. In addition, these genotypes may be useful in improving and breeding new restorer lines that are characterized by early flowering. The genotypes G19, G1, G14 and G4 recorded the lowest values of plant height, respectively. On the other hand, the genotypes G6, G8, G17, G18 and G20 showed the highest desirable values for the combined data over two years, as shown in Appendix A and Figure 2. Selection of new restorer lines of appropriate plant height (preferably 15–20 cm higher than CMS lines) with non-lodging characteristics is important for potential high-yielding hybrids [1,3,49]. The genotypes G14, G17 and G18 recorded the highest numbers of panicles/plant for both years and the combination of both years, as shown in Appendix A and Figure 3. These genotypes are desirable because this trait is closely correlated with a high grain yield per plant, [50,51]. Long panicles are usually correlated with a superior number of spikelets/panicle, resulting in a higher yield; therefore, a genotype with a longer panicle length is desirable [44,51,52]. Seven out of twenty-one studied genotypes recorded values higher than G1 (check variety). These genotypes included G9, G13, G3, G11, G18, G16 and G7, respectively, as shown in Appendix A and Figure 4. These genotypes are desirable for breeding promising lines. The panicle weight is correlated with grain yield; thus, most of these genotypes that showed the highest values of panicle weight also showed the highest values of grain yield [44,51,52,53]. The genotypes G11, G7, G20 and G9 gave the highest numbers of spikelets/panicle and number of filled grains/panicle, as shown in Appendix A and Figure 5 and Figure 6. The number of filled grains/panicle is one of the most important characteristics that directly affect the possibility of grain production in rice lines and their varieties [1,3,4,44,51,52]. In addition, 1000 grain weight is one of the most important traits that must be considered in hybrid rice breeding in Egypt [1,3]. The values of the studied genotypes were superior compared to the only commercially available restorer line regarding the 1000 g grain weight for the two-year and the combined data, as shown in Appendix A and Figure 7. These genotypes may be good restorer lines and useful in the development of new restorer lines with heavier grain that is more desirable for farmers in Egypt [1,3,4,44,51]. The genotypes G13, G11, G17, G15, G16, and G8 gave the highest values of grain yield/plant for the two-year and the combined data, as shown in Appendix A and Figure 8. These genotypes can be used as new restore lines and may be useful in the development of new restore lines in hybrid rice breeding programs in [1,3,4,12,44]. The genotypes G7, G15, G1, G12 and G10 showed the highest values of hulling percentage for the two-year and combined data, respectively, as shown in Appendix A. On the other hand, the genotypes G9, G14, G6, G17, G4, G3, G18, G15, G13, G7, and G10 recorded the highest values of milling percentage, as shown in Appendix A and Figure 9. Similar results were obtained by Dipti et al. (2003) [54]. For amylose content percentage, the genotypes G1, G2, G13, G11, and G8 showed desirable values that ranged between low or intermediate values for the combined data over two years, as shown in Appendix A. Similar results were obtained by [1,3,4,12]. The former genotypes can be used as new restorer lines to produce new rice hybrids with desirable amylose content and may also be useful in the development of new restore lines with suitable amylose content % for hybrid rice breeding programs in Egypt.

### 4.4. Analysis of Variances

The results of the analysis of variances showed the highly significant values of the mean squares of genotypes for the studied traits for the two years and combination of the two years, as shown in Table 2, Table 3, Table 4, Table 5, Table 6, Table 7 and Table 8. These findings indicated large differences among the rice genotypes for all traits in this study. Thus, the selection of better genotypes must be performed to improve the studied traits. These results agree with the results obtained by [1,3,4,55,56,57]. Moreover, the results of the analysis of variance for the combined data demonstrated that there were highly significant and significant differences between the years for all studied traits, except for panicle weight and 1000 g grain weight, as shown in Table 6, Table 7 and Table 8. In addition, the interaction between year and genotypes (year × genotypes) was highly significant for all studied traits, except days to 50% heading, 1000 g grain weight, grain yield/plant, hulling %, milling % and amylose content %. This result agrees with the result obtained by [1,2,3,58,59]. It can be concluded from the results of the analysis of variance that there is a wide variation in rice germplasm for all the quantitative and yield traits studied.

### 4.5. Genetic Parameters

The data in Table 9 show that for panicle weight, high values of phenotypic coefficients of variance were recorded. Moreover, the phenotypic coefficients and genotypic coefficients of variance were moderate for plant height, number of panicles/plants, number of spikelets per panicle, number of filled grains / panicle, grain yield per plant and amylose content percentage for the first-year, second-year and combined data, respectively, as shown in Table 9. The results of this study indicated the existence of exploitable genetic variability and diversity for these traits, and these results were in agreement with the results obtained by [1,2,3,60,61]. Therefore, phenotypic selection will be useful for the improvement of characteristics related to the degree of desired variance [1,2,60,61]. The relative values of genotypic and phenotypic coefficients of variation give essential information about the amount of variation. The estimates of genetic variability indicated that PCV values were slightly higher than those recorded for GCV that corresponded to the traits under study. These results indicate that phenotypic variation is largely determined by genotype; therefore, the selection of these traits on the basis of phenotype could be effective. Similar results were observed by [1,2,3,60,61]. It is clear from the results that the extent of the PCV% was higher than the coefficient of GCV% that corresponded to the studied traits. This can be explained by the influence of higher genotypes through environmental interaction and limited environmental influence on these traits, as suggested by [1,2,3,62]. 

The estimates of heritability in a broad sense (H2%) were high for all the studied traits, and the recorded values were more than 60% for the studied traits, except for the hulling percentage for the two-year and combined data (Table 9). Similar findings were also reported by [1,2,3,62,63,64,65,66]. Accordingly, it can be considered that the determination of the genotypes of nearly all the traits is associated with the determination of phenotypes. Thus, the selection of traits with high heritability values contributes to the improvement of traits that are less affected by the environment [1,3,4,62,66]. High estimates of heritability in a broad sense, and high genetic advance as a percentage of the mean were recorded for the following characteristics: PH, P/P, PW, Sp/P, Fg/P, GY/P and AC %, as shown in Table 9. These results indicate that these traits are governed by additive genes and that phenotypic selection is necessary to achieve the desirable improvement of these traits. Several previous studies obtained similar results to these results [67,68,69,70,71,72]. The presence of additive genetic variance provides a significant opportunity to exploit variance.

On the other hand, high heritability and moderate genetic advance were recorded for DH, PL, and 1000-GW. These results indicate that the 1000 grain weight, the number of days to 50% flowering and the panicle length were mostly inherited by the additive genes, and so phenotypic selection based on these traits may be beneficial. Similar results were reported by [63,69,70,73]. On the contrary, estimates of high heritability and low genetic advance were recorded for days to 50% flowering, PF%, SF%, and M%, as shown in Table 9. This result agrees with the results obtained by [1]. It also indicates that these traits are inherited by non-additive genes and that phenotypic selection is insufficient to achieve a desirable improvement in these traits. Using values of heritability that are associated with genetic advance would be better and more dependable than the use of heritability alone. It is for this reason that estimated heritability values alone are less dependable because these values are subject to change with changes in the environment and experimental material [63].

### 4.6. Cluster Analysis

A cluster that contained several varieties and lines was also obtained; these lines shared the same grain quality and high yielding potential. The dendrogram in Figure 10 also shows each group of closely related lines placed in a separate section. Twenty rice genotypes were identified in five clusters based on the fourteen characteristics, and the distribution of the genotypes was as follows: three genotypes were included in group C-A, six in C-B, six in C-C, one in C-D, and four in C-F, as shown in Table 10 and Figure 10. Clusters that contained a greater number of genotypes indicated a sufficient and large amount of genetic diversity [74,75,76]. As for the clusters that contained a smaller number of genotypes, this indicates the presence of a small amount of genetic diversity in the genotypes [75,77]. The cluster analysis grouped the 20 genotypes into 5 clusters (Table 10), including cluster B and C, which had same number of 6 genotypes for each, followed by cluster F, which had 4 genotypes, followed by cluster A, which had 2 genotypes, followed by cluster D, which had 1 genotype, indicating slight genetic diversity [75,78]. The lines and genotypes classified as cluster group D had higher mean values for seven of the traits studied. Cluster A had lower desirable mean values for two traits, cluster B had higher mean values for Sp/P and grain yield/plant and the cluster group F had higher mean values for three traits. The current study provides important information for selecting the optimum lines and genotypes for different traits. Therefore, this information is potentially useful and can also be used to generate greater variance in future rice breeding programs [78].

### 4.7. Principal Component Analysis

The separation of PC 1, PC 2, PC 3 and PC 4 demonstrated that 11 lines were detached in quarter 1 (G9, G18, G17, G16, G3 and G15) and quarter 2 (G14, G10, G5, G6, G4 and G2), indicating a high level of genotypic variation among these genotypes. The genotypes 9, 18, 17, 16, 3 and 15 and traits such as PW, PH, PL, PP, M%, 1000-GW, AC% and DH showed more variation than any other genotype or trait and were located in quarter 1, while SF% was located in quarter 2 (Figure 11 and Table 11). One of the most powerful and simple statistical techniques for reducing multivariate data is principal component analysis (PCA). PCA showed the same grouping pattern as the pattern observed in the cluster analysis, indicating that significant variation exists in this study.

Genotype x trait (GT) biplots provide a visualization of the true correlation between traits and an understanding of the existing relationships that facilitate the identification of traits that can be used in indirect selection for the improvement of grain yield. The results of the PCA of the agronomic characteristics of the genotypes studied showed that PC1 and PC2 were the best PCs in identifying the diversity between yield and its components, while PC4 was the optimal PC in identifying the diversity between grain quality traits (Figure 11 and Table 11). The traits in each group were collectively shaped by PCA. Based on the PCA results, PC1 traits included DH, P/P, PF%, SF%, 1000-GW and M% and these the groups had a higher percentage of variance. Moreover, from the PCA results, it was observed that the PC2 group included the traits PH, PL, PW, SP/P and GY/P and the groups had a higher proportion of variance. In addition, the results of the PCA revealed that the PC4 group included the traits H%, M% and AC% and these groups had a higher proportion of variance (Figure 11 and Table 11). These findings were in agreement with the findings from [79,80,81,82,83].

## 5. Conclusions

The main results of this study can be summarized as follows. The results of the cluster analysis showed that the studied genotypes can be grouped into five groups, including two groups labeled as B and C, each of which had six genotypes, followed by group F, which had four genotypes, and these results reflect the great genetic diversity in the groups with a larger number of genotypes. The lines and genotypes classified as cluster group D had higher mean values for seven of the traits studied. The lines and genotypes classified as cluster B had higher mean values for Sp/P and grain yield/plant. On the other hand, the lines and genotypes classified as cluster group F had higher mean values for three of the traits studied, including P/P, PL and H%. This study provides important information for selecting the optimum lines and genotypes for different traits. Therefore, this information is potentially useful and may also be used to generate greater variance in future rice breeding programs. Moreover, the results of the PCA of the traits of the genotypes studied showed that PC1 and PC2 were the best PCs in identifying the diversity between yield and its components, while PC4 was the optimal PC in identifying the diversity between grain quality traits. Furthermore, the most important outcomes of this study are the successful selection and identification of 17 new restoration lines of rice that can be used in rice and hybrid rice breeding programs.

## Figures and Tables

**Figure 1 genes-13-02227-f001:**
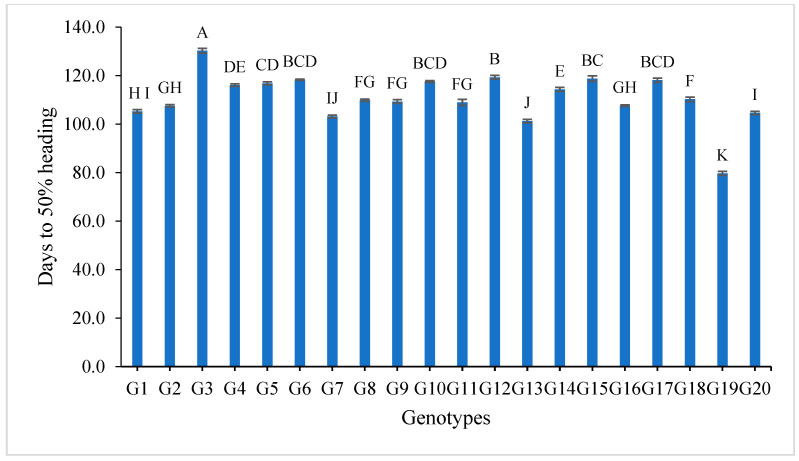
Mean values of days to 50% heading of twenty rice genotypes over the two years. Bar represents the standard deviation. Different alphabetic letters represent the significant differences among the genotypes at *p* < 0.05, based on Duncan’s multiple range test.

**Figure 2 genes-13-02227-f002:**
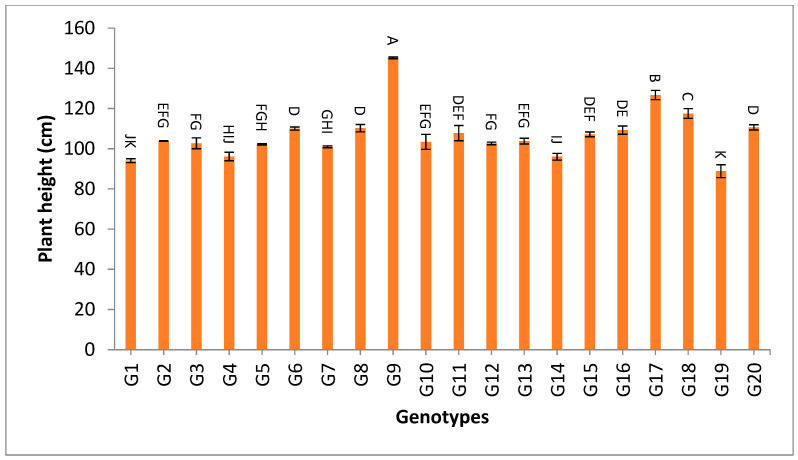
The mean values of plant height (cm) of twenty rice genotypes over the two years. Bar represents the standard deviation. Different alphabetic letters represent the significant differences among the genotypes at *p* < 0.05, based on Duncan’s multiple range test.

**Figure 3 genes-13-02227-f003:**
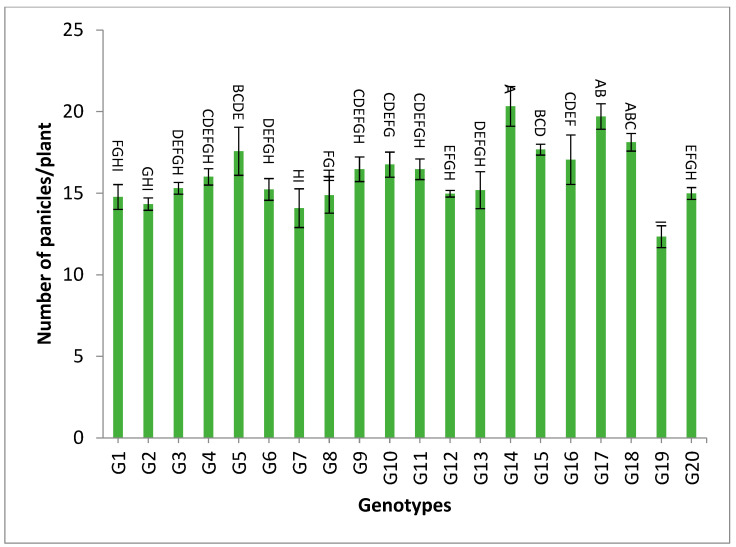
The mean numbers of panicles/plant of twenty rice genotypes over the two years. Bar represents the standard deviation. Different alphabetic letters represent the significant differences among the genotypes at *p* < 0.05, based on Duncan’s multiple range test.

**Figure 4 genes-13-02227-f004:**
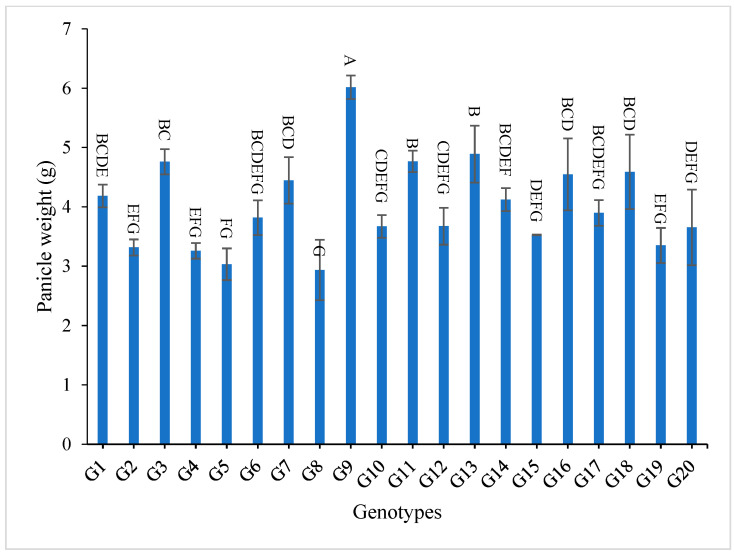
Mean values of panicle weight (g) of twenty rice genotypes over the two years. Bar represents the standard deviation. Different alphabetic letters represent the significant differences among the genotypes at *p* < 0.05, based on Duncan’s multiple range test.

**Figure 5 genes-13-02227-f005:**
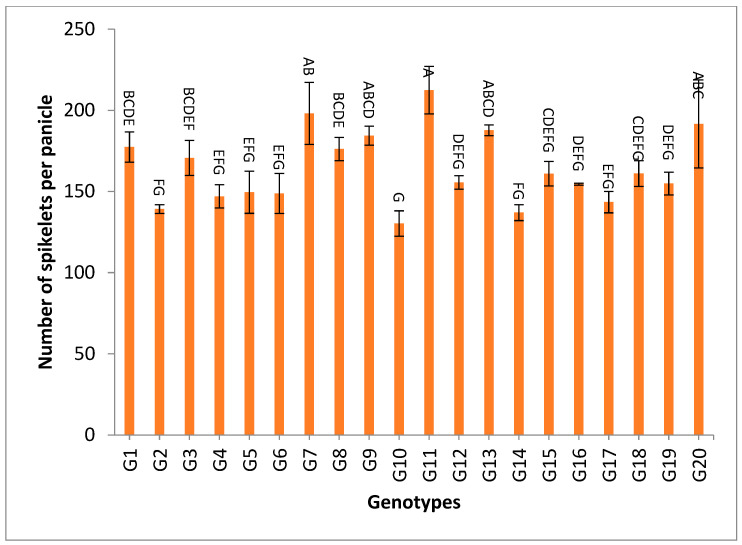
The mean numbers of spikelets per panicle of twenty rice genotypes over the two years. Bar represents the standard deviation. Different alphabetic letters represent the significant differences among the genotypes at *p* < 0.05, based on Duncan’s multiple range test.

**Figure 6 genes-13-02227-f006:**
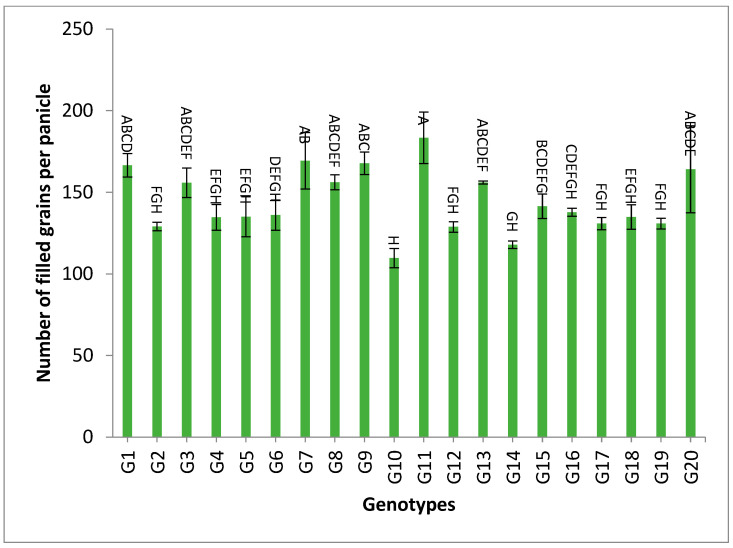
The mean numbers of filled grains per panicle of twenty rice genotypes over the two years. Bar represents the standard deviation.

**Figure 7 genes-13-02227-f007:**
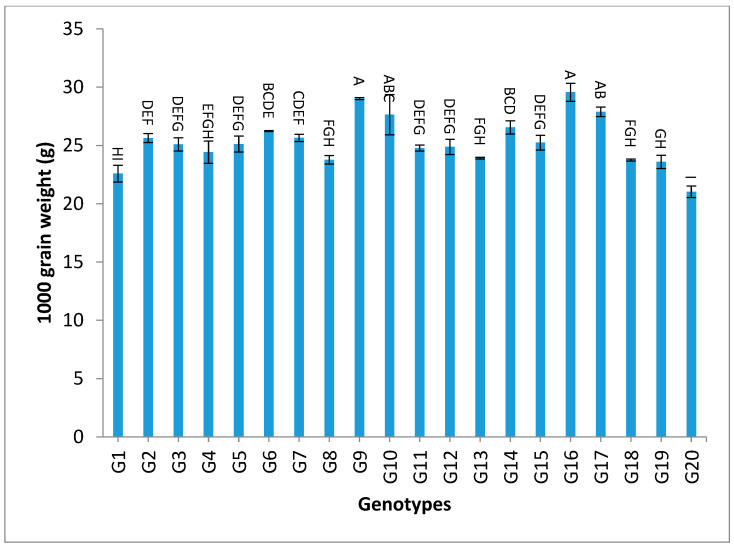
Mean values of 1000 grain weight (g) of twenty rice genotypes over the two years. Bar represents the standard deviation. Different alphabetic letters represent the significant differences among the genotypes at *p* < 0.05, based on Duncan’s multiple range test.

**Figure 8 genes-13-02227-f008:**
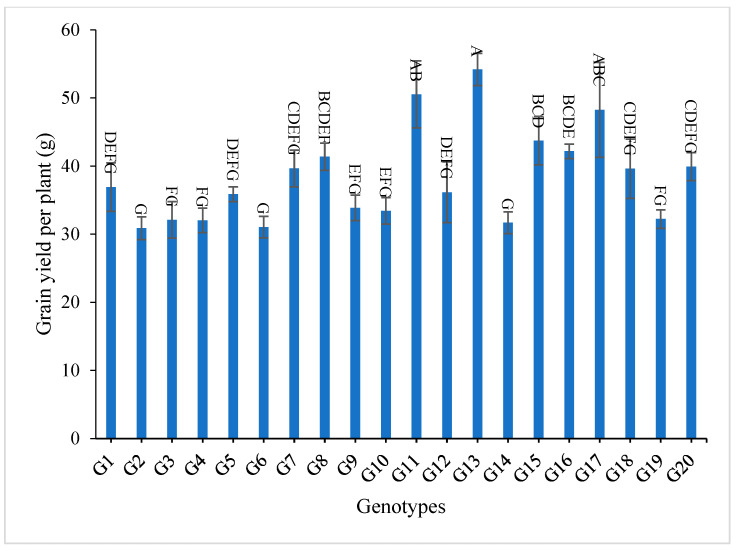
The mean values of grain yield/plant (g) of twenty rice genotypes over the two years. Bar represents the standard deviation. Different alphabetic letters represent the significant differences among the genotypes at *p* < 0.05, based on Duncan’s multiple range test.

**Figure 9 genes-13-02227-f009:**
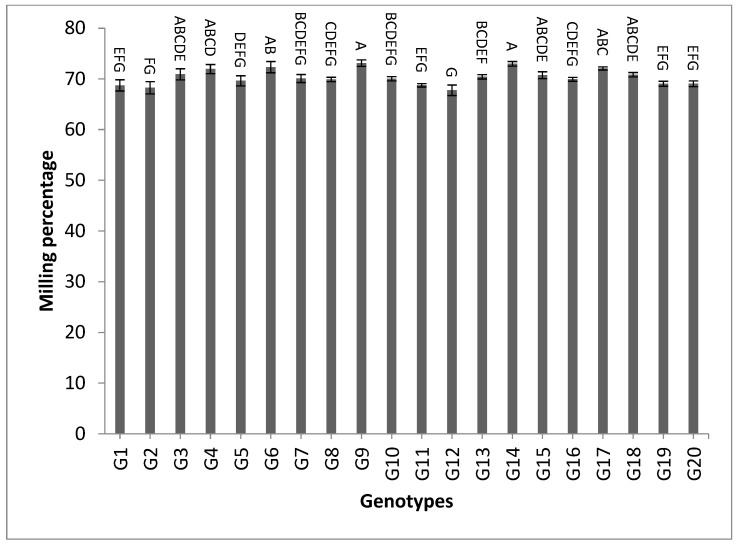
The mean values of milling percentage of twenty rice genotypes over the two years. Bar represents the standard deviation. Different alphabetic letters represent the significant differences among the genotypes at *p* < 0.05, based on Duncan’s multiple range test.

**Figure 10 genes-13-02227-f010:**
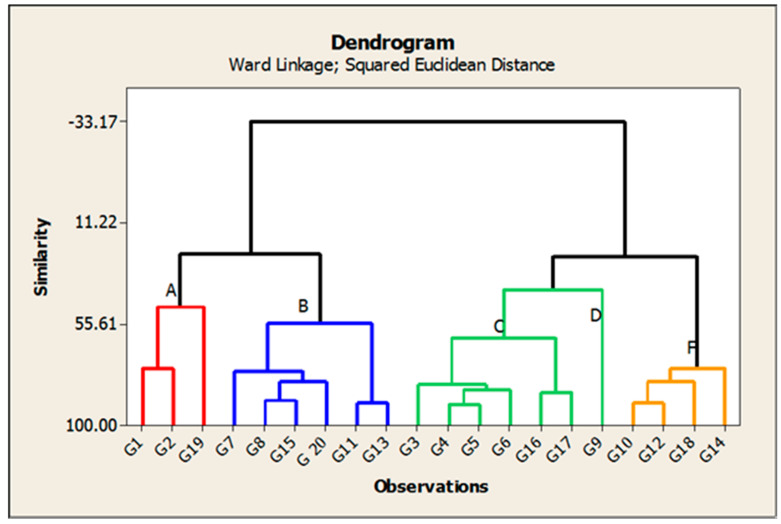
The dendrogram for the distance between the studied genotypes of rice based on yield and their components, as well as grain quality traits.

**Figure 11 genes-13-02227-f011:**
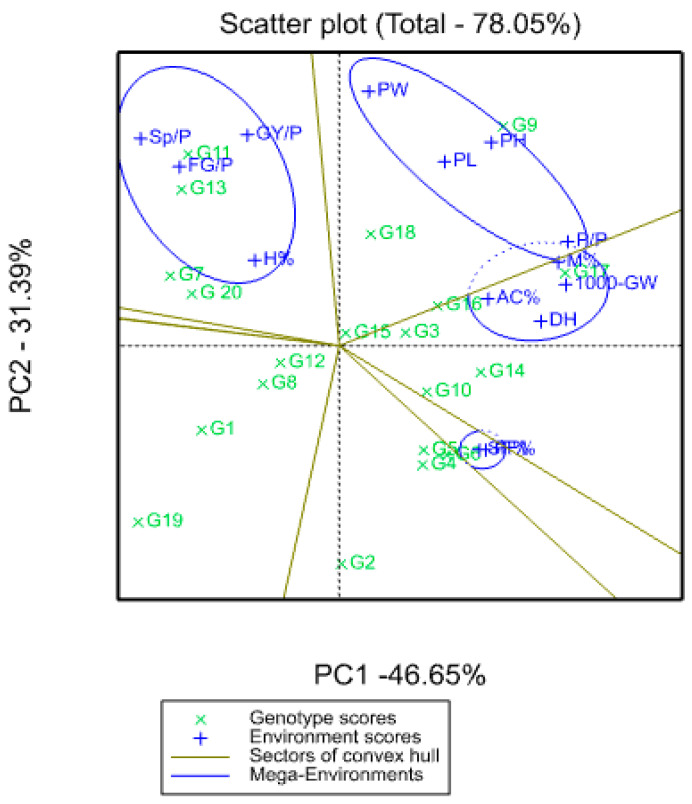
Genotype-by-trait (GT) biplot, showing the genotype-by-trait data.

**Table 1 genes-13-02227-t001:** List of rice genotypes used in this study.

ID	Name	Origin
G1	Giza 178	Egypt
G2	Giza 181	Egypt
G3	IR38	IRRI
G4	IR64	IRRI
G5	IR65	IRRI
G6	Tox 502-1-SLR1-LS3-B	Brazil
G7	Maybelle	USA
G8	H 156	China
G9	BR27	Bangladesh
G10	Xiangzi 3150	China
G11	IR71144-176-3-2-3-2	IRRI
G12	Oryzica 1-266-M	Colombia
G13	IR74052-184-3-3	IRRI
G14	IR72864-80	IRRI
G15	IR72868-11-1	IRRI
G16	PR2	India
G17	SPR85163	Thailand
G18	TOX 3133-59-1-2-4	IITA
G19	G46A,B	IRRI
G 20	IR79156A,B	IRRI

IRRI: International Rice Research Institute; IITA: International Institute of Tropical Agriculture.

**Table 2 genes-13-02227-t002:** Analysis of variances of days to 50% heading, plant height, number of panicles/plant and panicle length of twenty rice genotypes over two years.

		Mean Squares							
Source of Variance	d.f.	Days to 50% Heading		Plant Height (cm)		Number of Panicles/Plant		Panicle Length (cm)	
		2019	2020	2019	2020	2019	2020	2019	2020
Reps.	2	0.01	0.05	8.78	15.23	1.73	1.46	0.10	0.27
Genotypes	19	309.61 **	309.36 **	547.89 **	407.04 **	14.21 **	9.06 **	7.85 **	8.08 **
Error	38	0.88	0.45	4.10	7.74	0.85	0.79	0.86	1.05

** Highly significant at 1% level of probability.

**Table 3 genes-13-02227-t003:** Analysis of variances of panicle weight, pollen fertility percentage, number of spikelets /panicle and spikelet fertility percentage of twenty rice genotypes over two years.

		Mean Squares							
Source of Variance	d.f.	Panicle Weight (g)		Pollen Fertility (%)		Number of Spikelets/Panicle		Spikelet Fertility (%)	
		2019	2020	2019	2020	2019	2020	2019	2020
Reps.	2	0.20	0.11	1.15	2.46	20.2	2.82	1.15	2.46
Genotypes	19	1.61 **	2.30 **	63.10 **	35.59 **	1872.98 **	2208.71 **	63.33 **	34.95 **
Error	38	0.21	0.14	4.28	3.84	197.91	132.29	4.28	3.84

** Highly significant at 1% level of probability.

**Table 4 genes-13-02227-t004:** Analysis of variances of number of filled grains/panicle, 1000 g grain weight and grain yield/plant of twenty rice genotypes over two years.

		Mean Squares					
Source of Variance	d.f.	Number of Filled Grains/Panicle		1000 g Grain Weight (g)		Grain Yield/Plant (g)	
		2019	2020	2019	2020	2019	2020
Reps.	2	21.12	1.55	0.33	0.09	6.44	6.44
Genotypes	19	1286.26 **	1674.52 **	13.53 **	13.43 **	145.59 **	133.34 **
Error	38	157.20	94.42	1.09	0.49	9.54	9.54

** Highly significant at 1% level of probability.

**Table 5 genes-13-02227-t005:** Analysis of variances of grain quality of twenty rice genotypes over two years.

		Mean Squares					
Source of Variance	d.f.	Hulling Percentage		Milling Percentage		Amylase Content %	
		2019	2020	2019	2020	2019	2020
Reps.	2	0.12	0.12	0.22	0.22	0.13	0.13
Genotypes	19	3.36 **	2.82 **	7.65 **	6.85 **	41.83 **	39.66 **
Error	38	0.71	0.71	0.59	0.59	0.61	0.61

** Highly significant at 1% level of probability.

**Table 6 genes-13-02227-t006:** The combined analysis of yield and its components of twenty rice genotypes over the two years.

		Mean Squares					
Source of Variance	d.f.	Days to 50% Heading	Plant Height (cm)	Number of Panicles/Plant	Panicle Length (cm)	Panicle Weight (g)	Pollen Fertility (%)
Year (Y)	1	7.65 **	1860.39 **	47.38 **	18.33 **	0.01	169.39 **
Error	4	0.026	12.01	1.6	0.19	0.155	1.81
Genotypes (G)	19	618.96 **	905.58 **	21.49 **	13.20 **	3.48 **	72.09 **
Y × G	19	0.008	49.36 **	1.78 **	2.73 **	0.43 **	26.60 **
Error	76	0.665	5.92	0.82	0.96	0.175	4.06

** Highly significant at 1% level of probability.

**Table 7 genes-13-02227-t007:** The combined analysis of yield and its components of twenty rice genotypes over the two years.

		Mean Squares				
Source of Variance	d.f.	Number of Spikelets/Panicle	Spikelet Fertility (%)	Number of Filled Grains/Panicle	1000 g Grain Weight (g)	Grain Yield/Plant (g)
Year (Y)	1	253.75 **	169.39 **	64.59 *	0.31	241.97 **
Error	4	11.51	1.81	11.33	0.21	6.44
Genotypes (G)	19	3007.83 **	71.68 **	2250.11 **	26.40 **	278.61 **
Y × G	19	1073.85 **	26.60 **	710.67 **	0.57	0.32
Error	76	165.10	4.06	125.81	0.79	9.54

** Highly significant at 1% level of probability, * significant at 5% level of probability.

**Table 8 genes-13-02227-t008:** The combined analysis of grain quality traits of twenty rice genotypes over the two years.

		Mean Squares		
Source of Variance	d.f.	Hulling Percentage	Milling Percentage	Amylase Content %
Year (Y)	1	30.88 **	30.88 **	28.78 **
Error	4	0.12	0.22	0.13
Genotypes (G)	19	6.00 **	14.33 **	81.24 **
Y × G	19	0.18	0.18	0.25
Error	76	0.71	0.59	0.62

** Highly significant at 1% level of probability.

**Table 9 genes-13-02227-t009:** The genetic parameters and broad sense heritability of studied traits for studied rice genotypes for the two-year and combined data.

Traits	Genotypic Coefficient of Variation (G.C.V %)			Phenotypic Coefficient of Variation (P.C.V %)			Heritability in a Broad Sense			Genetic Advance in Percentage (Expected)		
	2019	2020	Combined	2019	2020	Combined	2019	2020	Combined	2019	2020	Combined
Days to 50% heading	9.17	9.13	9.16	9.21	9.15	9.18	99.15	99.57	99.43	18.82	18.78	18.81
Plant height (cm)	12.14	11.20	11.44	12.28	11.52	11.58	97.79	94.51	97.49	24.37	22.42	23.26
Number of panicles/plant	12.65	10.77	13.09	13.81	12.21	14.71	83.92	77.79	79.15	23.88	19.57	21.70
Panicle length (cm)	6.01	6.21	5.64	7.03	7.48	6.46	72.96	69.00	76.05	10.57	10.63	10.13
Panicle weight (g)	16.99	21.12	18.24	20.39	23.12	20.22	69.41	83.39	81.38	29.16	39.73	33.9
Pollen fertility (%)	4.98	3.56	3.74	5.50	4.16	4.05	82.07	73.39	85.17	9.29	6.29	7.11
Number of spikelets per panicle	14.28	16.18	13.09	16.62	17.66	14.71	73.83	83.95	79.15	25.27	30.54	23.98
Spikelet fertility (%)	5.10	3.60	3.81	5.63	4.22	4.13	82.12	72.99	85.09	9.52	6.34	7.24
Number of filled grains/panicle	13.51	15.82	12.79	16.09	17.18	14.59	70.54	84.80	76.88	23.38	30.01	23.1
1000 g grain weight (g)	8.03	8.22	8.15	9.02	8.67	8.54	79.11	89.79	81.38	14.71	16.04	16.02
Grain yield per plant (g)	18.27	16.18	17.18	20.10	17.96	18.98	82.62	81.22	81.93	30.04	34.21	32.04
Hulling percentage	1.19	1.05	1.10	1.59	1.48	1.52	55.59	49.95	51.99	1.82	1.52	1.63
Milling percentage	2.20	2.04	2.11	2.46	2.31	2.37	80.10	78.11	78.94	4.05	3.72	3.86
Amylase content %	15.61	14.60	15.07	15.96	14.94	15.42	95.72	95.49	95.59	31.47	29.38	30.36

**Table 10 genes-13-02227-t010:** Summary of cluster analysis of the twenty genotypes of rice.

Cluster ID	No. of Genotypes	GEN	DH	PH	P/P	PL	PW	PF%	Sp/P	SF%	FG/P	1000-GW	GY/P	H%	M%	AC%
A	3	G1, G2, G19	97.5	95.6	13.8	23.	3.62	92.3	157.2	90.4	142.2	23.9	33.3	79.5	68.7	20.30
B	6	G7, G8, G15, G20, G11, G13	107.7	106.8	15.3	25.	4.04	88.0	187.8	86.2	161.8	24.1	44.9	80.2	69.8	22.74
C	6	G3, G4, G5, G6, G16, G17	117.9	107.9	16.8	25.2	3.88	93	152.3	90.9	138.4	26.4	36.9	79.1	71.1	26.01
D	1	G9	109.3	145.2	16.5	25.5	6.01	93.2	184.4	91.3	167.8	29.0	33.9	80.1	73.1	28.96
F	4	G10, G12, G18, G14	115.4	104.9	17.5	26.2	4.01	86.5	146	84.5	122.8	25.7	35.2	80.3	70.4	25.55

**Table 11 genes-13-02227-t011:** The principal component analysis of measured characteristics of twenty genotypes of rice.

Variable	PC1	PC2	PC3	PC4
DH	0.331	0.044	−0.018	−0.21
PH	0.25	0.364	−0.154	0.093
P/P	0.375	0.186	0.169	−0.214
PL	0.173	0.329	0.233	−0.438
PW	0.049	0.455	−0.16	0.199
PF%	0.24	−0.186	−0.526	−0.054
Sp/P	−0.326	0.371	−0.243	0.041
SF%	0.23	−0.186	−0.535	−0.049
FG/P	−0.262	0.32	−0.401	0.036
1000-GW	0.372	0.109	−0.008	0.125
GY/P	−0.15	0.377	0.017	−0.386
H%	−0.138	0.153	0.16	0.502
M%	0.36	0.15	−0.046	0.304
AC%	0.244	0.084	0.237	0.39

## Data Availability

Not applicable.

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
