# Peer review of "Utilization of Genetic Resources, Genetic Diversity and Genetic Variability for Selecting New Restorer Lines of Rice (Oryza sativa L.)"

_genes, 2022, doi:10.3390/genes13122227_

Round 1

Reviewer 1 Report

Major comments:

I shall suggest extensive English improvement of the article. I found several mistakes regarding sentence structure, grammar and proper use of tenses.

Introduction section is too long and has so many short paragraphs with redundancy in information. I shall suggest that please merge the paragraphs with similar information and make not more than 3-4 standard sized paragraphs for better presentation of your objectives, background and available literature.

I shall suggest to check the genetic diversity of your germplasm using trait specific molecular markers and present their result in your MS.

I shall suggest to add the error bars on the graphs particularly Figure1-9.

Your tables are showing that genotypes are significantly different from each other in different parameters. I shall suggest that please run the post-hoc test on your significant data and present the results on appropriate place.

Minor comments:

L3: Replace “select” with “selecting

L26-27:  Please consider rephrasing the sentence. I think you should remove “, heredity” then it could be quite meaningful.

L30: Write “Twenty rice genotypes …” instead of “The materials were 20 rice genotypes”.

L35: It should be “plant height”, instead of “height plant”.

L38: Please remove the space between 60 and %. It should be like 60%. Consider this comment throughout the manuscript (MS).

L51: Write “world” instead of “world's”.

L71: Please write the full form of any abbreviation at its first use e.g., IRRI. Consider this comment throughout the MS.

L73: Please correct “world. [13].”

L94: Remove “comma” from the text “group, [18].”

L117-131, 157-166: Short paragraphs and so many grammatical mistakes. Please correct.

L195: “The plant material is 20 rice genotypes selected …” can be like “Twenty (20) rice genotypes were selected …”

L197: “2” should be written as “two”. Under “10” the digits should be written in words and 10 and above can be written in digits. But, in the start of paragraphs the numbers should be written in words in any case. Please check it throughout the MS.

Table 1: why “TOX 3133-59-1-2-4” is red. If there is any specific information related to it then please give the legends.

L195-255: So many English mistakes and short paragraphs. Please check and correct.

L558: Probably it is Figure 10. Please check.

Author Response

Response to the Review Report (Reviewer 1)

Comments and Suggestions for Authors

Major comments:

I shall suggest extensive English improvement of the article. I found several mistakes regarding sentence structure, grammar and proper use of tenses.

Introduction section is too long and has so many short paragraphs with redundancy in information. I shall suggest that please merge the paragraphs with similar information and make not more than 3-4 standard sized paragraphs for better presentation of your objectives, background and available literature.

I shall suggest to check the genetic diversity of your germplasm using trait specific molecular markers and present their result in your MS.

Sorry, We did not have the capabilities to perform molecular markers.

I shall suggest to add the error bars on the graphs particularly Figure1-9. Done

Your tables are showing that genotypes are significantly different from each other in different parameters. I shall suggest that please run the post-hoc test on your significant data and present the results on appropriate place.

Minor comments:

L3: Replace “select” with “selecting Done

L26-27:  Please consider rephrasing the sentence. I think you should remove “, heredity” then it could be quite meaningful. Done

L30: Write “Twenty rice genotypes …” instead of “The materials were 20 rice genotypes”. Done

L35: It should be “plant height”, instead of “height plant”. Done

L38: Please remove the space between 60 and %. It should be like 60%. Consider this comment throughout the manuscript (MS). Done

L51: Write “world” instead of “world's”. Done

L71: Please write the full form of any abbreviation at its first use e.g., IRRI. Consider this comment throughout the MS. Done

L73: Please correct “world. [13].” Done

L94: Remove “comma” from the text “group, [18].” Done

L117-131, 157-166: Short paragraphs and so many grammatical mistakes. Please correct. Done

L195: “The plant material is 20 rice genotypes selected …” can be like “Twenty (20) rice genotypes were selected …” Done

L197: “2” should be written as “two”. Under “10” the digits should be written in words and 10 and above can be written in digits. But, in the start of paragraphs the numbers should be written in words in any case. Please check it throughout the MS. Done

Table 1: why “TOX 3133-59-1-2-4” is red. If there is any specific information related to it then please give the legends.

"TOX 3133-59-1-2-4" in red. To remember writing what the abbreviation IITA stands for

L195-255: So many English mistakes and short paragraphs. Please check and correct. Done

L558: Probably it is Figure 10. Please check. Done

Reviewer 2 Report

1. (Number of spikelets per panicle) Figure 5 and Figure 4 (panicle weight) are the same?

2. Why is there no statistical analysis for each graph? None of them have a vertical coordinate? 

3. The figures and tables in the article are not standard, you can combine the figures.

4. Why do you choose these 20 varieties?

5. Do these rice restorer lines differ greatly in their resilience? What are the advantages?

6. Are there any differences at the gene level in the new restorer lines developed by the authors? Have you done molecular labeling?

Author Response

Response to the Review Report (Reviewer 2)

Comments and Suggestions for Authors

  1. (Number of spikelets per panicle) Figure 5 and Figure 4 (panicle weight) are the same? Corrected
  2. Why is there no statistical analysis for each graph? None of them have a vertical coordinate? Done
  3. The figures and tables in the article are not standard, you can combine the figures. Done
  4. Why do you choose these 20 varieties?

The studied genotypes, i.e., lines and varieties were selected of germplasm available for the rice, then evaluated in the following year, observed, and crossed with the two most important cytoplasmic male sterile lines was done to set up the experimental test-cross. Based on the percentage of pollen fertility and the percentage of spikelet fertility of the resulting hybrids and the results of the test crosses experiment, the selected lines were divided based on the results of these two traits of the resulting hybrids and their genetic behavior to maintainer lines, partial maintainer, partial restorer and restorer lines. In the next step, the best 17 of the restorer fertility lines were selected for cultivation and evaluated with the two maintainer lines and the only restorer fertility commercial line. The eighteen restorer lines were also re-crossed with the same two cytoplasmic male sterile lines,

  1. Do these rice restorer lines differ greatly in their resilience? What are the advantages?

These rice recovery lines differ greatly among themselves, and these selected lines are distinguished in their superiority over the commercial restorer line, as they are more synchronized in flowering with the cytoplasmic male sterile lines (CMS), and many of them are better than the commercial restorer line in plant height. The grain size and grain shape of them, and that most of it is superior to the commercial restorer line in the grain yield and is also superior to it in the quality characteristics studied.

  1. Are there any differences at the gene level in the new restorer lines developed by the authors? Have you done molecular labeling?

The selected restorer lines under this study differ in the country of origin, as they are from distant and different regions, and also differ in their phenotypic characteristics. Also, these restorer lines differ in the characteristics of the hybrids resulting from them by crossing with two the cytoplasmic male sterile lines (CMS). Sorry, We did not have the capabilities to perform molecular labeling.

Round 2

Reviewer 1 Report

Dear authors,

Thank you very much for improving your MS. Nevertheless, the improvements are not satisfactory. Introduction is still problematic. Methods can be improved. In results section, error bars on all graphs are surely not correct. Moreover, results and discussion should be presented separately and clearly. Results can be better presented through structure analysis. English language must be improved and revised. Significant errors found in writing style. For example, there are several short paragraphs or paragraphs with inappropriate length. I shall suggest that you must work on all these areas before your next subission.

Author Response

Author's Reply to the Review Report (Reviewer 1)

Thank you very much for efforts.

Comments and Suggestions for Authors

Dear authors,

Thank you very much for improving your MS. Nevertheless, the improvements are not satisfactory. Introduction is still problematic. Done

Methods can be improved. Done

In results section, error bars on all graphs are surely not correct. Done

Moreover, results and discussion should be presented separately and clearly. Results can be better presented through structure analysis. English language must be improved and revised. Significant errors found in writing style. For example, there are several short paragraphs or paragraphs with inappropriate length. I shall suggest that you must work on all these areas before your next subission. Done

Reviewer 2 Report

The manuscript could be accepted to be published in Gene.

Author Response

Author's Reply to the Review Report (Reviewer 2)

Thank you very much for your efforts.

Round 3

Reviewer 1 Report

Still the paragraphs in introduction are short and more in number. Better to merge the paragraphs with similar story (this is the same comment that I made in my first report).

The titles of Table 2, 3 & 4 are 100% same. If the information is same then why two tables? and if the information is different, which I think so, then why the same titles.

ANOVA tables show that the traits were highly significant then I shall suggest that please present the results of post-hoc test and put letters, to show significance, on the mean values of the genotypes in the graph charts.

Table 10 is showing certain abbreviations. Have you presented their full form anywhere in the article? Please check

Please merge 2nd and 3rd paragraphs of heading “4.3. Genetic parameters”.

To support your hypothesis and results, please remove old and irrelevant references.

Still English language improvement is needed. For example:

Line 544 – Rephrase the sentence.

Line 557 – Remove “,” placed after “to”.

Line 570-571 – Rephrase the sentence.

Line 590 – Citation error. Please check.

Line 597-98 – Rephrase the sentence.

Author Response

Still the paragraphs in introduction are short and more in number. Better to merge the paragraphs with similar story (this is the same comment that I made in my first report).

Done, its changed

The titles of Table 2, 3 & 4 are 100% same. If the information is same then why two tables? and if the information is different, which I think so, then why the same titles. Done, its changed

Table 2. Analysis of variances of yield and its components of twenty rice genotypes in two years

Changed to :

Table 2. Analysis of variances of days to 50% heading, plant height, number of panicles/plant and panicle length of twenty rice genotypes in two years

Table 3. Analysis of variances of yield and its components of twenty rice genotypes in two years

Changed to :

Table 3. Analysis of variances of panicle weight, pollen fertility percentage, number of spikelets /panicle and spikelets fertility percentage of twenty rice genotypes in two years

Table 4. Analysis of variances of yield and its components of twenty rice genotypes in two years

Changed to :

Table 4. Analysis of variances of number of filled grains / panicle, 1000-grain weight and grain yield/plant of twenty rice genotypes in two years

ANOVA tables show that the traits were highly significant then I shall suggest that please present the results of post-hoc test and put letters, to show significance, on the mean values of the genotypes in the graph charts. Done

Table 10 is showing certain abbreviations. Have you presented their full form anywhere in the article? Please check

Done, line 710, 930, and 934

Please merge 2nd and 3rd paragraphs of heading “4.3. Genetic parameters”. Done

To support your hypothesis and results, please remove old and irrelevant references.

Done

Still English language improvement is needed. For example:

Line 544 – Rephrase the sentence. Done

Line 557 – Remove “,” placed after “to”.  I can't see it, please delete it

Line 570-571 – Rephrase the sentence. The line has no paragraph

Line 590 – Citation error. Please check. The line has no paragraph

Line 597-98 – Rephrase the sentence Done
